**Data Availability Statement:** All relevant data are within the paper and its Supporting information files.

**Funding:** Yin Yen-Liang Foundation Development and Construction Plan" of the School of Medicine,

# Key pathological features characterize minimal change disease-like IgA nephropathy

**Tsung-Yueh Wang[1], Fu-Pang Chang[2,3], An-Hang Yang[2,3,4], Shuk-Man Ka[5], Ann Chen[6], Jyh-Tong Hsieh[1], Fan-Yu Chen[1], Tsung-Lun Lee[1], Po-Yu Tseng[2,7], Ming-Tsun Tsai[1,2], Szu-Yuan Li[1,2], Chih-Yu Yang** ◉ [1,2,4,8,9,10]*, **Jinn-Yang Chen[1,2], Chih-Ching Lin[1,2], Der-Cherng Tarng[1,2,4,8]**

**1** Department of Medicine, Division of NephrologyTaipei Veterans General Hospital, Taipei, Taiwan, **2** Faculty of Medicine, School of Medicine, National Yang Ming Chiao Tung University, Taipei, Taiwan, **3** Department of Pathology, Taipei Veterans General Hospital, Taipei, Taiwan, **4** Institute of Clinical Medicine, School of Medicine, National Yang Ming Chiao Tung University, Taipei, Taiwan, **5** Graduate Institute of Life Sciences, National Defense Medical Center, Taipei, Taiwan, **6** Department of Pathology, Tri-Service General Hospital, National Defense Medical Center, Taipei, Taiwan, **7** Department of Medicine, Division of Nephrology, Taipei City Hospital, Heping Fuyou Branch, Taipei, Taiwan, **8** Center for Intelligent Drug Systems and Smart Bio-devices (IDS2B), Hsinchu, Taiwan, **9** Stem Cell Research Center, National Yang Ming Chiao Tung University, Taipei, Taiwan, **10** Department of Medicine, Division of Clinical Toxicology and Occupational Medicine, Taipei Veterans General Hospital, Taipei, Taiwan

* cyyang3@vghtpe.gov.tw

## Abstract

### Aims

A subset of IgA nephropathy (IgAN) patients exhibiting minimal change disease (MCD) like features present with nephrotic-range proteinuria and warrants immunosuppressive therapy (IST). However, the diagnosis of MCD-like IgAN varied by reports. We aimed to identify the key pathological features of MCD-like IgAN.

### Methods

In this cohort, 228 patients had biopsy-proven IgAN from 2009 to 2021, of which 44 without segmental sclerosis were enrolled. Patients were classified into segmental (< 50% glomerular capillary loop involvement) or global (> 50%) foot process effacement (FPE) groups. We further stratified them according to the usage of immunosuppressant therapy after biopsy. Clinical manifestations, treatment response, and renal outcome were compared.

### Results

26 cases (59.1%) were classified as segmental FPE group and 18 cases (40.9%) as global FPE group. The global FPE group had more severe proteinuria (11.48 [2.60, 15.29] vs. 0.97 [0.14, 1.67] g/g, $p = 0.001$) and had a higher proportion of complete remission (81.8% vs. 20%, $p = 0.018$). In the global FPE group, patients without IST experienced more rapid downward eGFR change than the IST-treated population (-0.38 [-1.24, 0.06] vs. 1.26 [-0.17, 3.20]mL/min/1.73 m$^2$/month, $p = 0.004$).

National Yang-Ming University, Taipei, Taiwan (107F-M01-0504, 107F-M01-0510, and 111Q58502Y), the National Science and Technology Council (NSTC), Taiwan (MOST109-2314-B-010-053-MY3, MOST 110-2811-B-010-510, MOST 110-2813-C-A49A-551-B, and MOST 110-2321-B-A49-003, NSTC 112-2314-B-A49-059-MY3), grants from Taipei Veterans General Hospital, Taipei, Taiwan (V111C-155, V111D63-003-MY2, and VGHUST111-G6-7-2), and the "Center for Intelligent Drug Systems and Smart Bio-devices (IDS2B)" from The Featured Areas Research Center Program within the framework of the Higher Education Sprout Project by the Ministry of Education (MOE) in Taiwan. The funders have no role in study design, data collection, analysis, interpretation, or manuscript writing.

**Competing interests:** The authors have declared that no competing interests exist.

## Conclusions

The absence of segmental sclerosis and the presence of global FPE are valuable pathological features that assist in identifying MCD-like IgAN.

## Introduction

The incidence of IgA nephropathy (IgAN) is estimated at 2.5 per 100,000 person-year and is considered the most common primary immune-mediated glomerulonephritis [1]. Asymptomatic hematuria is a common manifestation of IgAN. Concomitant minimal proteinuria may be discovered. Nephrotic-range proteinuria, gross hematuria with coincident pharyngitis, and rapid renal function decline are less common [2]. Nephrotic syndrome occurs in 5% of IgAN patients[3]. IgAN with nephrotic syndrome is considered to be a variant form of IgAN and can have electron microscopic features resembling minimal change disease (MCD). The treatment strategy in this MCD-like variant form is in accordance with MCD rather than IgAN [4]. In a combined cohort study, the 10-year renal survival of IgAN was estimated at 57–94%, and histologic features like segmental sclerosis and interstitial fibrosis were thought to be poor prognostic factors [5]. In a retrospective cohort, the renal outcome in steroid-resistant IgAN patients with nephrotic syndrome variant was not more favorable than that of classic IgAN [6].

A consensus pathological classification of IgAN, Oxford classification, was published in 2009 [7]. Oxford classification variables include mesangial hypercellularity (M), endocapillary hypercellularity (E), segmental sclerosis (S), and tubular atrophy/interstitial fibrosis (T), also known as the MEST score. M, S, and T scores are independently predictive of renal function decline rate and survival from renal failure. The E score has renal predictive value in patients without immunosuppressant therapy (IST) [8].

Although IgAN is characterized by predominant IgA deposition in the mesangium, podocytopathy may present. Focal segmental glomerulosclerosis (FSGS)-like or MCD-like features can be observed upon IgAN pathological examination. The presence of segmental glomerulosclerosis is proved to be a poor prognostic factor for proteinuria and renal survival in IgAN patients [9]. By contrast, the absence of segmental sclerosis is a key diagnostic feature in MCD [10]. However, the identification of MCD-like IgAN varied by study. A retrospective study of 17 cases of IgAN with MCD required the absence of endocapillary proliferation for enrolled cases [11]. Another prospective cohort of 27 cases of steroid use in MCD-like IgAN patients utilized diffuse podocyte FPE and electron-dense material deposition in mesangium as inclusion criteria [12]. To our knowledge, no generally acknowledged pathological diagnosis of MCD in the context of IgAN has been proposed. We postulate that the absence of segmental sclerosis and extensive FPE would best characterize MCD-like IgAN.

## Materials and methods

### Study subjects

In this study, we retrospectively reviewed the pathological report and clinical information of biopsy-proven IgAN patients at our institute from November 2009 to February 2021. Patients aged more than 18 years old were eligible. The diagnostic feature of MCD includes the absence of segmental sclerosis but the absence of endocapillary hypercellularity is not included [10]. Therefore, S = 0 by the Oxford classification is required. Cases with transplant kidneys or

other concomitant pathological diagnoses such as FSGS or diabetic nephropathy were excluded.

The protocol of this study was approved by the Institutional Review Board of Taipei Veterans General Hospital, Taipei, Taiwan (IRB-TPEVGH No.: 2018-06-011AC). The protocol conformed with the ethical guidelines of the Helsinki Declaration. The need for informed consent was waived because of the retrospective nature of the study.

## Pathological and clinical characterization

For light microscopic examination, all specimens were fixed in buffered formalin and embedded in paraffin, and stained with hematoxylin and eosin, Masson trichrome, periodic acid-Schiff, and silver stains. For immunofluorescence staining, all frozen samples were stained with fluorescein isothiocyanate conjugated rabbit antisera to human IgG, IgA, IgM, C1q, C3, C4, kappa, and lambda light chains. The mesangial IgA immunofluorescence (IF) stain intensity was graded as 0, trace, 1 to 3+. The IgA immunostaining intensity was required to be at least 1+ in our enrolled patients. For electron microscopy examination, resin-embedded samples were prepared. Mesangial electron-dense deposits and the severity of foot process effacement were assessed. As shown in Fig 1, the severity of FPE was categorized as segmental if < 50% of glomerular capillary loop involvement or as global if > 50% of that. The whole glomeruli in each specimen for electron microscopic analysis were examined thoroughly. The patients were classified into segmental or global FPE groups accordingly. The individual pathology characteristics were shown in S1 Table. Pathological features, including Oxford classification variables and immunostaining intensity of IgA and C3, were compared between segmental and global FPE groups. The detailed staining protocol was described in supporting information.

The proteinuria was quantified by the urine protein/creatinine ratio (UPCR) of the single-voided urine sample. The severity of hematuria was graded from 0 to 4+. The estimated glomerular filtration rate (eGFR) was calculated by the CKD-EPI equation [13]. Laboratory assessment of serum creatinine and albumin within 1 week before the biopsy was recorded as initial eGFR and albumin. The presence of lower limbs edema or not was acquired from the admission note. Blood pressure was record on the first day of admission, or outpatient department visit within 1 month before admission. Body mass index(BMI) was calculated from height and weight recorded on the day of admission. Nephrotic range proteinuria was defined as UPCR >3.5 g/g. Nephrotic syndrome was defined as nephrotic range proteinuria, serum albumin < 2.5 g/dL, and presence of edema. Clinical features, including hematuria, initial albumin, initial eGFR, BMI, blood pressure, nephrotic syndrome, and the presence of edema, were compared between the segmental and global FPE groups. As for proteinuria severity comparison, patients with prior IST or renin-angiotensin-aldosterone system (RAAS) blockade were further excluded. The remaining cases were further classified according to IST use or not. The IST regimen was either steroid alone, steroid in combination with cyclophosphamide, or steroid in combination with mycophenolate mofetil. Global FPE with IST, global FPE without IST, segmental FPE with IST, and segmental FPE without IST were designated as group 1, group 2, group 3, and group 4 respectively.

## Treatment response and renal outcome

The response to IST was defined as complete remission (CR) if UPCR < 0.3 g/g after treatment, partial remission (PR) if proteinuria reduction > 50% but UPCR > 0.3 g/g, and no response (NR) if proteinuria reduction < 50%. The follow-up UPCR was assessed approximately 3 months after initiation of IST. The IST regimen and RAAS blockade of the individual

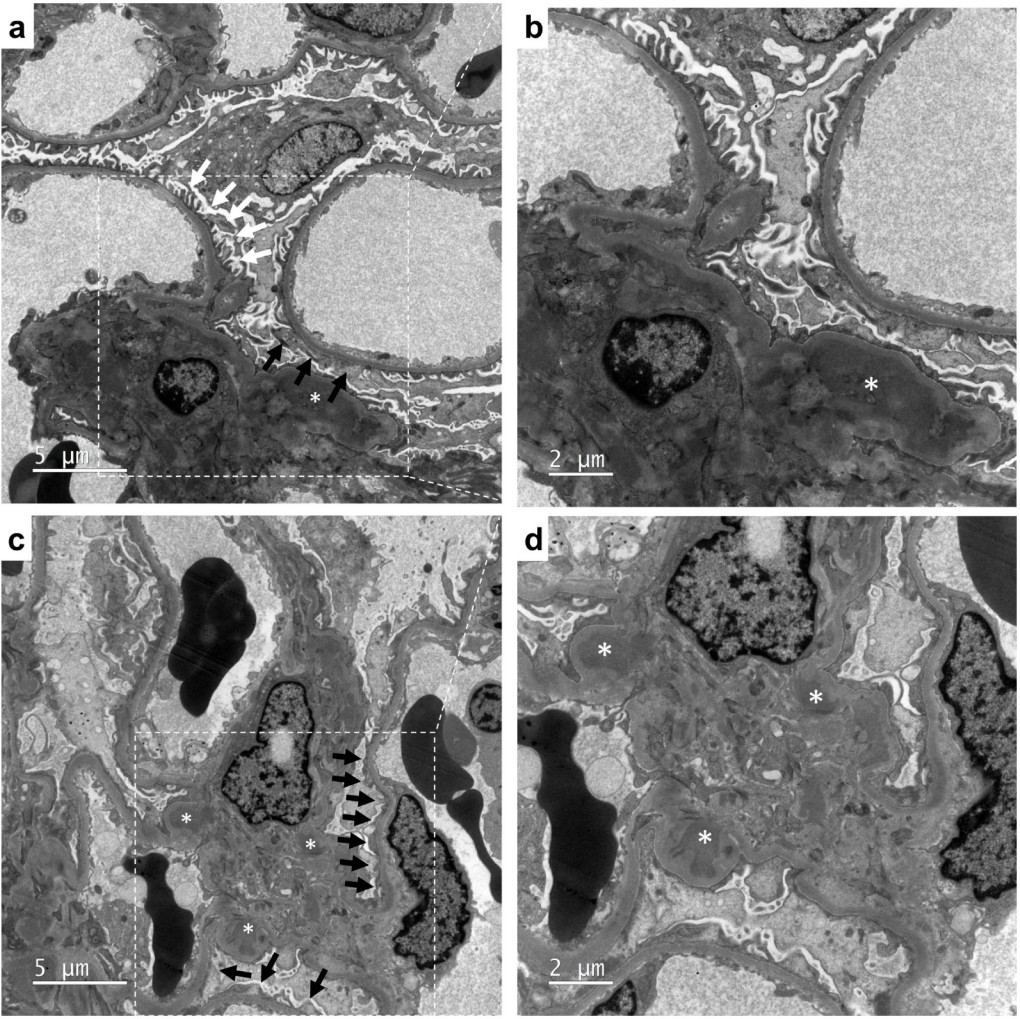

**Fig 1. Representative kidney pathology to classify global and segmental foot process effacement (FPE) by electron microscopy.** The segmental FPE is defined as < 50% of glomerular capillary loop involvement, as shown in panels (a) (6,000× magnification) and (b) (10,000× magnification). In contrast, the global FPE is defined as > 50% of glomerular capillary loop involvement. In panels (c) (6,000× magnification) and (d) (10,000× magnification), FPE is extensive, and no normal part was seen. Black arrows denote FPE, white arrows denote normal podocyte foot processes, and asterisks denote mesangial electron-dense deposits.

patient are listed in S2 Table. The eGFR change rate was estimated by the slope of the least square regression linear model, with a positive value representing improvement and a negative value deterioration. The time frame of the eGFR change rate assessment was within two years after the biopsy. The renal endpoint was defined as a > 10% decline in eGFR. The treatment response and eGFR change rate were compared between different patient groups.

## Statistical analysis

Continuous variables were expressed as mean ± standard deviation (SD) if normally distributed and median with interquartile range [IQR; $25^{th}$ quartile, $75^{th}$ quartile] if non-normally distributed. The categorical variables were expressed as percentages. The *student's* t-test was used to compare normally distributed data, the Mann-Whitney U test for non-normally

distributed data, and the Chi-square test for categorical variables. Kaplan-Meier method was used to depict the survival curve between groups, and the Log-rank test was used to analyze the difference between groups. All probabilities were two-tailed, and a *p*-value of less than 0.05 was statistically significant. All statistical analyses were performed using SPSS Software ver. 19.0 (Armonk, NY: IBM Corp.).

## Results

### Patient enrollment

A total of 228 patients had biopsy-proven IgAN. As shown in Fig 2, one sample was from a graft kidney, six patients were under 18 years old at the time of biopsy, and 23 patients had other concomitant pathological diagnoses. Eighteen cases had inadequate specimens for the Oxford classification interpretation and 12 inadequate specimens for electron microscopic (EM) examination. Two cases were previously diagnosed with ankylosing spondylitis, one had a history of systemic lupus erythematosus (SLE), and another one had a history of Sjogren syndrome. These 4 cases were excluded in consideration of secondary IgAN. Forty-four of the remaining cases had no history of liver cirrhosis, inflammatory bowel disease, lung cancer, lymphoma, human immunodeficiency virus infection, or autoimmune diseases. They were classified as the absence of segmental sclerosis (S = 0), of which 26 cases (59.1%) were identified as segmental effacement and 18 cases (40.9%) as global effacement.

### Baseline characteristics

The cohort enrolled 21 males and 23 females. The mean age of the patients was 43.7 ± 17.7 years old at the time of the biopsy. Five (11.3%) had prior RAAS blockade use, and none had prior IST use. Two of the enrolled patients had diabetes mellitus. The mean systolic blood pressure was 132.8 ± 20.3 mmHg and the mean diastolic blood pressure was 77.4 ± 13.5

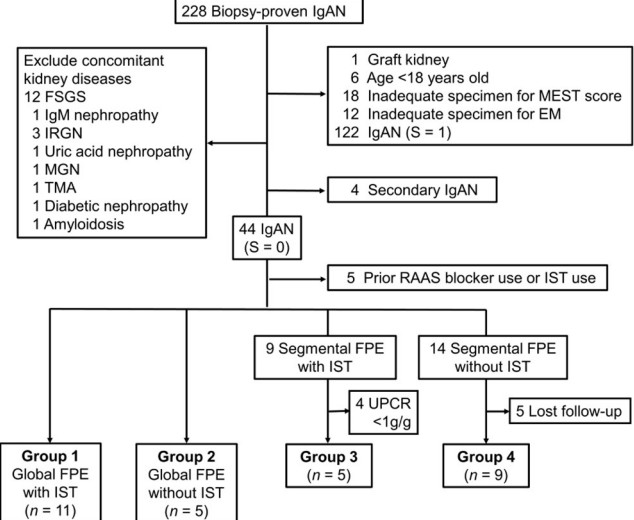

**Fig 2. Flow chart of the study.** Abbreviations: IgAN, IgA nephropathy; FSGS, focal segmental glomerulosclerosis, IRGN, infection-related glomerulonephritis; MGN, membranous glomerulonephritis; TMA, thrombotic microangiopathy; MEST, mesangial hypercellularity, endocapillary hypercellularity, segmental sclerosis, and interstitial fibrosis/tubular atrophy; EM, electron microscopy; RAAS, renin-angiotensin-aldosterone system; IST, immunosuppressant therapy; FPE, foot process effacement; UPCR, urine protein/creatinine ratio.

**Table 1. Demographic characteristics and pathological features.**

| | Segmental FPE | Global FPE | p-Value |
|---|---|---|---|
| **Patient number (n; %)** | 26; 59.1 | 18; 40.9 | |
| **Age** | 42.5 ± 16.4 | 45.4 ± 19.7 | 0.598 |
| **Male gender (%)** | 11; 42.3 | 10; 55.6 | 0.387 |
| **Prior RAAS blockade use (%)** | 3; 11.5 | 2; 11.1 | 0.965 |
| **Prior IST (%)** | 0: 0.0 | 0; 0.0 | 1.000 |
| **Albumin (g/dL)** | 3.8 ± 0.54 | 2.65 ± 1.17 | 0.001 |
| **Edema (n; %)** | 2; 7.7 | 12; 66.7 | 0.001 |
| **Nephrotic syndrome (n; %)** | 0; 0% | 11; 61.1% | 0.001 |
| **Hematuria (%)[†]** | | | 0.017* |
| **4+ (n; %)** | 2; 8.3 | 0; 0.0 | |
| **3+ (n; %)** | 3; 12.5 | 0; 0.0 | |
| **2+ (n; %)** | 8; 33.3 | 1; 5.6 | |
| **1+ (n; %)** | 5; 20.8* | 11; 61.1* | |
| **0 (n; %)** | 6; 25.0 | 6; 33.3 | |
| **Initial eGFR, mL/min/1.73m²** | 77.2 ± 36.8 | 77.4 ± 38.7 | 0.985 |
| **Systolic blood pressure, mmHg[††]** | 137 ± 21.8 | 127.2 ± 17.2 | 0.121 |
| **Diastolic blood pressure, mmHg[††]** | 79.57 ± 14.3 | 74.65 ± 12.2 | 0.252 |
| **BMI, kg/m²** | 24.6 ± 3.9 | 26.4 ± 4.8 | 0.218 |
| **MEST score** | | | |
| **M = 0 (n; %)** | 13: 50.0 | 14; 77.8 | 0.063 |
| **E = 0 (n; %)** | 23; 88.5 | 18; 100.0 | 0.135 |
| **S = 0 (n; %)** | 26; 100.0 | 18; 100.0 | 1.000 |
| **T = 0 (n; %)** | 23; 88.5 | 16; 88.9 | 0.965 |
| **IgA stain intensity** | | | 0.559 |
| **1+ (n; %)** | 2; 7.7 | 3; 16.7 | |
| **2+ (n; %)** | 9; 34.6 | 7; 38.9 | |
| **3+ (n; %)** | 15; 57.7 | 8; 44.4 | |
| **C3 stain intensity** | | | 0.117 |
| **0 (n; %)** | 3; 11.5% | 2; 11.1% | |
| **Trace (n; %)** | 6; 23.1% | 4; 22.2% | |
| **1+ (n; %)** | 5; 19.2% | 9; 50% | |
| **2+ (n; %)** | 12; 46.2% | 3; 16.7% | |
| **3+ (n; %)** | 0; 0% | 0; 0% | |

Data were expressed as mean ± SD or number; percentage as appropriate.

*p < 0.05.

[†]Urine RBC data were not available in two cases of the segmental group before the biopsy.

[††]Blood pressure data were not available in four cases.

Abbreviations: FPE, foot process effacement; RAAS, renin-angiotensin-aldosterone system; IST, immunosuppressant therapy; eGFR, estimated glomerular filtration rate; MEST hypercellularity, mesangial hypercellularity, endocapillary hypercellularity, segmental sclerosis, and interstitial fibrosis/tubular atrophy.

mmHg. The mean BMI was 25.1 ± 4 kg/m². Fourteen (31.8%) of the enrolled patients had nephrotic range proteinuria. Eleven (25%) of the enrolled patients presented with nephrotic syndrome. C3 immunostaining positivity was found in thirty-nine (88.6%) patients. The clinical characteristics and pathological feature differences between the segmental and global FPE groups were summarized in Table 1. The level of serum albumin was significantly lower in the global FPE group (2.65 ± 1.17 vs. 3.8 ± 0.54 g/dL, *p* = 0.001). The presence of edema prevailed

**Table 2. Proteinuria difference between segmental/global FPE groups.**

|  | Segmental FPE | Global FPE | p-Value |
|---|---|---|---|
| **n** | 23 | 16 |  |
| **Initial UPCR (g/g)** [§] | 0.97 [0.14, 1.67] | 11.48 [2.60, 15.29] | 0.001 |

Data were expressed as median [IQR].

[§]Patients with prior RAAS blockade usage or IST had been excluded.

Abbreviations: FPE, foot process effacement; UPCR, urine protein/creatinine ratio; RAAS, renin-angiotensin-aldosterone system; IST, immunosuppressant therapy.

in the global FPE group (66.7% vs. 7.7%, $p = 0.001$). The age, gender, eGFR, prior RAAS blockade, and IST use had no significant statistical difference between groups. The hematuria differs in less severe grading of 1+ ($p = 0.017$). The individual proportion distribution of MEST variables and immunostaining intensity of C3, IgA were not significantly different between groups.

For comparing differences in the severity of proteinuria, patients with prior RAAS blockade or IST ($n = 5$) were excluded. The group with global FPE had more prominent proteinuria than the segmental FPE group (11.48 [2.60, 15.29] vs. 0.97 [0.14, 1.67] g/g, $p = 0.001$), as shown in Table 2.

## Treatment response and renal outcome

After excluding RAAS blockade users and prior IST-treated patients, the remaining 39 patients were stratified by global or segmental FPE, post-biopsy IST users, or non-users to four groups, as shown in Fig 2. Treatment response discrepancies were compared between groups 1 and 3. The likelihood of CR was significantly higher in the global FPE group (81.8% vs. 20%, $p = 0.018$), as illustrated in Fig 3. In both groups 1 and 3, the time from biopsy to IST was within 3 months, as shown in Table 3. The clinical characteristics according to immunosuppression therapy responsiveness are listed in S3 Table.

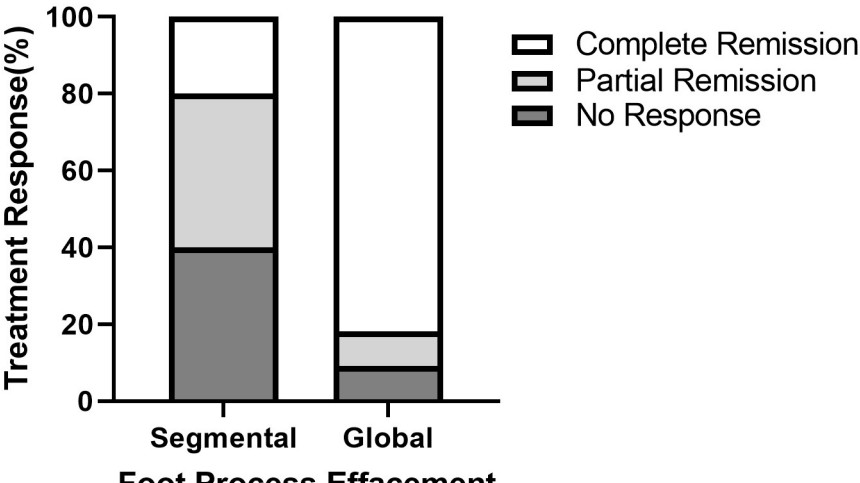

**Fig 3. Treatment response differences between segmental and global foot process effacement groups.** Complete remission was achieved in 81.8% of the global FPE group and 20% in the segmental FPE group ($p = 0.018$).

**Table 3. The renal outcome comparison between different patient groups.**

|  | Group | Patient number | eGFR decline rate, mL/min/1.73m²/month | Follow-up duration, months | Time to IST, months |
|---|---|---|---|---|---|
| **Global FPE IST(+)** | 1 | 11 | 1.26 [0.17, 3.20] | 13.4 [9.7, 18.8] | 0.07 [0.00, 0.57] |
| **Global FPE IST(−)** | 2 | 5 | -0.38 [-1.24, 0.06] | 12.5 [6.0, 18.7] | N/A |
| **Segmental FPE IST(+)** | 3 | 5 | 0.31 [-0.71, 0.80] | 13.1 [7.1, 23.3] | 0.33 [0.25, 0.6] |
| **Segmental FPE IST(−)** | 4 | 9 | -0.52 [-1.72, 0.47] | 8.2 [2.4, 12.8] | N/A |

Data were expressed as median [IQR]. Abbreviations: eGFR, estimated glomerular filtration rate; IST, immunosuppressant therapy; FPE, foot process effacement; N/A: Not applicable.

eGFR change rate was compared between groups 1 and 2, groups 3 and 4, groups 1 and 3, and groups 2 and 4, as shown in Table 3. Significant statistical difference was discovered between groups 1 and 2 (1.26 [0.17, 3.20] vs. -0.38 [-1.24, -0.06] mL/min/1.73 m²/month, $p = 0.004$), but not in groups 3 and 4 (0.31 [-0.71, 0.80] vs. -0.52 [-1.72, 0.47] mL/min/1.73 m²/month, $p = 0.205$). The Kaplan-Meier curve delineated the survival analysis of renal outcomes between groups, and the difference was significant between groups 1 and 2 as examined by the Log-rank test ($p = 0.039$), as shown in Fig 4. In contrast, there is no difference between groups 3 and 4 ($p = 0.627$). Clinical characteristics including RAAS blockade use after biopsy, blood pressure, IST duration, and BMI of each group were summarized in Table 4. The value of BMI of

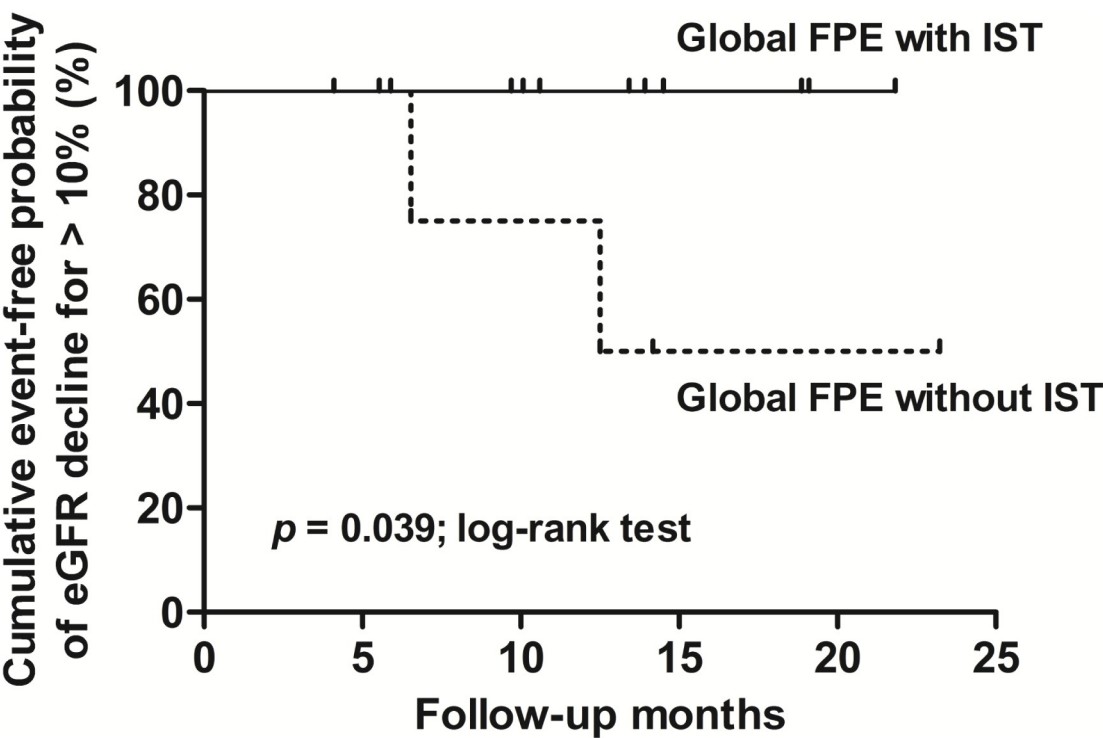

**Fig 4. Kaplan-Meier curve for renal event-free survival comparison between groups 1 and 2 (p = 0.039)** Abbreviations: FPE, foot process effacement; IST, immunosuppressant therapy.

**Table 4. Clinical characteristics of different groups.**

| | Group 1 (n = 11) | Group 2 (n = 5) | Group 3 (n = 5) | Group 4 (n = 9) | p value |
|---|---|---|---|---|---|
| **RAAS blockade (n; %)** | 4; 36.4 | 0; 0 | 1; 20 | 4; 44 | 0.322 |
| **SBP, mmHg** | 121[119, 136] | 120[113, 133] | 135[115, 153] | 127[110, 134] | 0.709 |
| **DBP, mmHg** | 81[69, 84] | 73[58, 89] | 76[65, 82] | 73[63, 90] | 0.986 |
| **Duration of IST, months** | 6[3, 18] | N/A | 26[7, 44] | N/A | 0.101 |
| **BMI, kg/m$^2$** | 27.4[25.4, 29.6] | 22.5[21.2, 25.7] | 22.2[19.5, 27.4] | 23[22.2, 26.6] | 0.036* |

Data were expressed as median [IQR].

*Statistically significant between group 1 and group 2.

Abbreviations; IST, Immunosuppressant therapy; RAAS, renin-angiotensin-aldosterone system; SBP, systolic blood pressure; DBP, diastolic blood pressure; BMI, body mass index; N/A: Not applicable.

group 1 was statistically higher than that of group 2 (27.4[25.4, 29.6] vs. 22.5[21.2, 25.7] kg/m$^2$, $p$ = 0.036). None of the other clinical characteristics had statistical significance between groups.

## Discussion

In this study, patients with global FPE presented with a less severe degree of microscopic hematuria, were more often to have nephrotic-range proteinuria and had an excellent response to IST compared with the segmental FPE group. The two indices, absence of segmental sclerosis(S = 0) and global FPE were valuable features identifying MCD-like IgAN.

Absence of endocapillary hypercellularity(E = 0) was not required in our cohort. Though all of the patient with global FPE happened to be classified as E = 0, the proportion had no significant difference to that of the segmental group. Additionally, one drawback of endocapillary hypercellularity scoring is that its reproducibility was poorer than segmental sclerosis scoring [7]. Counting macrophage numbers in CD68-stained section of glomerulus may be an appropriate alternative for endocapillary hypercellularity scoring [14].

Tubulointerstitial fibrosis (T = 1 or 2) is frequently associated with hypertension in adults, particularly in the elderly, and does not necessarily exclude MCD [15]. So it was not listed as an exclusion criteria in our study. In a multi-center retrospective cohort, patients with pathological change T1 were found to be more resistant to steroid [16]. In our cohort, three patients were resistant to steroid in treatment group and two of them were pathologically classified as T = 1. By contrast, none of the steroid-responsive IgAN patients were pathologically classified as T = 1 or 2.

One review article stated that nephrotic syndrome was seen in 5% of IgAN patients [3]. In our cohort, sixteen patients were classified as global FPE and accounted for 7% of 228 biopsy-proven IgAN patients. Eleven patients were classified as having nephrotic syndrome and the percentage was 4.8%. One Korean cohort study reported that 48 among 581 IgAN patients (8.3%) presented with nephrotic syndrome [17]. However, the hypoalbuminemia criteria in this Korean cohort was extended to be lower than 3.5mg/dL, which might account for higher prevalence of nephrotic syndrome in IgAN patients.

KDIGO guidance recommends steroid treatment in IgAN with MCD-like features [4]. However, the recommendation lack valid evidence because of the rare population, and the related studies enrolled IgAN patients with nephrotic syndrome rather than IgAN with MCD-like pathological features [17–19]. In our study, IST in MCD-like IgAN patients attenuated the eGFR decline rate and had better renal outcomes than the non-IST-treated group. These patients mostly achieved CR, and the time to IST was within 3 months. Only one patient

classified as MCD-like IgAN was defined as NR in treatment response. Whether the resistant response to first-line glucocorticoid treatment in MCD-like IgAN patients suggest poor renal outcomes needs further investigation. In condition of MCD, a substantial portion of patients progressed to ESKD if unresponsive to glucocorticoid treatment [20].

In IgAN patients classified as segmental FPE (i.e., not MCD-like IgAN) in our cohort, though the eGFR change rate appeared to deteriorate faster in the non-IST treated group than in IST-treated group, it did not reach statistical significance (-0.52 [-1.72, 0.47] vs. 0.31 [-0.71, 0.80] mL/min/1.73 m$^2$/month, $p$ = 0.205). In an Italian randomized control trial, ninety-seven IgAN patients were allocated to receive oral steroid therapy plus RAAS blockade use or RAAS blockade only. The median follow-up duration was 5 years. The mean rate of eGFR decline was higher in the RAAS blockade-only group (-6.17 ± 13.3 vs. −0.56 ± 7.62 mL/min/1.73 m2/ year; $p$ = 0.013) [21]. No such difference was observed in our study. Inconsistent, relatively short follow-up duration and small sample size in our cohort and higher initial dose of oral prednisolone (1mg/kg/day) in the Italian study may contribute to this difference.

Whether the extensive foot process effacement observed in MCD-like IgAN should be interpreted as podocytopathic variant of IgAN or coexistence of MCD remained unknown. The podocyte reacts and adapts to mechanical or immunological stress, and FPE protects against these injuries [22]. The FPE observed in obesity-related FSGS appeared to be segmental (< 50% of the glomerular capillary surface involvement) compared with idiopathic FSGS [23]. Mechanical shear force by glomerular hyperfiltration stresses the slit diaphragm and podocyte unevenly, thus resulting in segmental FPE [24]. In contrast, extensive FPE is more often seen in direct podocyte injury-like virus-associated FSGS or innate cytoskeletal derangement like primary FSGS [25]. Lupus podocytopathy, characterized by the absence of immune complex deposition but extensive FPE, was a particular subclass of the population recognized in recent years [26]. The devoid of the immune complex implied other mechanisms of podocyte cyto-toxicity mediated by cytokines or lymphocyte dysregulation [27]. As mentioned above, the immune complex-mediated or mechanical stress-induced FPE, these adaptive changes tend to cause segmental rather than the extensive form of FPE. The extensive foot process effacement is more likely to be the result of podocyte cytotoxicity.

The pathogenesis of IgAN is initiated by the formation of circulating galactose-deficient IgA (Gd IgA). Antibodies against these aberrantly-glycosylated IgA constituted an immune complex. The tendency of Gd IgA to aggregate with other Gd-IgA molecules or IgG may facili-tate the process of forming an immune complex [28]. The deposition of the immune complex trigger mesangial cells to release humoral cytokines and hence the resultant podocytopathy [29]. As discussed in previous paragraph, the immune complex-mediated FPE tend to be seg-mental. The global FPE observed in MCD-like IgAN may be more appropriately considered as coexistence of MCD. However, more investigations are needed to address this issue.

Our study had several limitations. First, the follow-up duration was relatively short consid-ering the indolent nature of IgA or MCD, and a more significant renal function decline differ-ence may be observed if the follow-up duration was longer. Second, an assessment of daily proteinuria should be done with 24-hour urine collection. Third, sample size was small, owing to low prevalence of the IgAN with MCD. Lastly, different IST regimens were implemented in the treatment groups.

## Conclusion

The absence of segmental sclerosis and the presence of global FPE are valuable pathological features identifying MCD-like IgAN. Prompt IST ameliorates the severity of proteinuria effec-tively and halts renal function decline rate in this population.

## Supporting information

**S1 Table. Individual renal biopsy pathology characteristics.**
(PDF)

**S2 Table. Individual IST and RAAS blockade regimen.**
(PDF)

**S3 Table. Individual clinical characteristics according to steroid response.**
(PDF)

**S1 Dataset. Baseline characterisrtics of each enrolled patient.**
(XLSX)

**S2 Dataset. Renal function change and treatment response in each group.**
(XLSX)

**S1 File. Staining protocol.**
(PDF)

## Acknowledgments

We thank Dr. Pu-Yuan Chang (Institute of Clinical Medicine, School of Medicine, National Yang Ming Chiao Tung University, Taipei, Taiwan) for his critical reading of the manuscript and stimulating discussions during the preparation of this article.

## Author Contributions

**Conceptualization:** Tsung-Yueh Wang, An-Hang Yang, Shuk-Man Ka, Ann Chen, Chih-Yu Yang.

**Data curation:** Tsung-Yueh Wang, Jyh-Tong Hsieh, Fan-Yu Chen, Tsung-Lun Lee.

**Formal analysis:** Tsung-Yueh Wang, Chih-Yu Yang.

**Funding acquisition:** Chih-Yu Yang.

**Investigation:** Fu-Pang Chang, An-Hang Yang, Shuk-Man Ka, Ann Chen, Po-Yu Tseng, Ming-Tsun Tsai, Szu-Yuan Li, Chih-Yu Yang, Jinn-Yang Chen, Chih-Ching Lin, Der-Cherng Tarng.

**Methodology:** Tsung-Yueh Wang, Fu-Pang Chang, Chih-Yu Yang.

**Project administration:** Tsung-Yueh Wang, Fu-Pang Chang, Chih-Yu Yang.

**Supervision:** Chih-Yu Yang, Chih-Ching Lin, Der-Cherng Tarng.

**Visualization:** Tsung-Yueh Wang, Chih-Yu Yang.

**Writing – original draft:** Tsung-Yueh Wang, Jyh-Tong Hsieh, Fan-Yu Chen, Tsung-Lun Lee, Chih-Yu Yang.

**Writing – review & editing:** Tsung-Yueh Wang, Fu-Pang Chang, An-Hang Yang, Shuk-Man Ka, Ann Chen, Po-Yu Tseng, Ming-Tsun Tsai, Szu-Yuan Li, Chih-Yu Yang, Jinn-Yang Chen, Chih-Ching Lin, Der-Cherng Tarng.

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
