## [Decision Letter · Decision Letter 0]

22 Nov 2022

PONE-D-22-24589Key pathological features characterize minimal change disease-like IgA nephropathyPLOS ONE

Dear Dr. Yang,

Thank you for submitting your manuscript to PLOS ONE. After careful consideration, we feel that it has merit but does not fully meet PLOS ONE’s publication criteria as it currently stands. Therefore, we invite you to submit a revised version of the manuscript that addresses the points raised during the review process.**The manuscript focuses on a topic of potential interest. The study, however, presents several major shortcomings that should be addressed. To mention some of them, i) need to temper the conclusion in the abstract, as the study has limitations, especially the small sample size; ii) unclear whether a definition of MCD like IgAN would require also mesangial IgA deposition as one of the hallmarks of IgAN; iii) need to provide all staining protocols; iv) need to define in the methods which clinical and pathological features are compared in the study; v) need to describe in detail the type of immunosuppression applied, and describe in detail the treatment (duration, in combination with RAS blockade etc); vi) need to define over which time frame is the decline of eGFR; vii) need to mention in the Results baseline characterizes; viii) need to clarify if cases with secondary IgAN have been excluded and if not, how they are distributed in the groups; ix) need to provide some information about gd-IgA to relate to the clinical course (remission and non-remission) as well as global FPE and segmental FPE; x) unclear why patients with tubulointerstitial fibrosis as sign of potentially advanced IgAN with concomitant proteinuria have not been excluded; xi) need to clarify on which basis the remission criteria were chosen; xii) need to explain the high percentage of MCD^+^ IgAN (approx. 8%) in the cohort compared to other studies; xiii) need to explain the eGFR decline rate of 1.17 ml/months in the segmental, non-IST group; xiv) consider a detailed discussion on the treatment before the authors can conclude that resistance to IST treatment may suggest poor renal outcome; xv) need to clarify  the discussion on adaptive changes of the podocyte and to focus on data in IgAN, xvi) need to include the serum albumin levels of the patients in this cohort and the presence/absence of edema to enable better interpretation of the data.**

We look forward to receiving your revised manuscript.

Kind regards,

Giuseppe Remuzzi

Academic Editor

PLOS ONE

https://journals.plos.org/plosone/s/fileid=ba62/PLOSOne_formatting_sample_title_authors_affiliations.pdf.

Reviewers' comments:

Reviewer's Responses to Questions

**Comments to the Author**

1. Is the manuscript technically sound, and do the data support the conclusions?

Reviewer #1: Partly

Reviewer #2: Yes

2. Has the statistical analysis been performed appropriately and rigorously? 

Reviewer #1: Yes

Reviewer #2: Yes

3. Have the authors made all data underlying the findings in their manuscript fully available?

Reviewer #1: No

Reviewer #2: Yes

4. Is the manuscript presented in an intelligible fashion and written in standard English?

Reviewer #1: Yes

Reviewer #2: Yes

5. Review Comments to the Author

Reviewer #1: Wang et al report the results of an observational study aiming to evaluate key pathological features of MCD-like IgAN. The topic is relevant to further differentiate MCD like IgAN. The authors evaluated a cohort of 228 patients with biopsy proven IgAN.

The strength of the study is the effort made to further determine MCD like IgAN. Limitations are related to the small sample size, missing clinical data and definitions of treatment periods and an inconsistent discussion of the data.

I think that the study might be published only if an extensive revision of the manuscript has been performed to achieve a valid contribution to the scientific record.

Comments to the authors:

Abstract:

1) be careful with a too confident conclusion, as the study has limitations, especially sample size.

Introduction:

2) Proteinuria is not “frequently minimal” in IgAN and macrohematuria after upper airway infection is a classic and common in IgAN.

3) IgA is deposited in the Mesangium rather than in the mesangial cell

4) Wouldn’t a definition of MCD like IgAN require also Mesangial IgA deposition as one of the hallmarks of IgAN?

Material and Methods:

1) All staining protocols are missing, essential for reproduction of data. C3 staining is not mentioned in the methods and is found only in the supplement. IgA antibodies and procedures for IF are not described etc. They should be provided.

2) Define in the methods which clinical and pathological features are compared in the study.

3) Describe in detail the type of immunosuppression applied (steroid or other?) and describe in detail the treatment (duration, in combination with RAS Blockade etc.) Has RAS Blockade been applied in patients without IST (in all patients after diagnosis)?

4) Define the decline of eGFR: over which time frame?

Results:

1) Baseline characteristics: SBP and DBP, percentage of NS, C3 deposition, Diabetes mellitus (if present), BMI should be mentioned.

2) Please clarify if cases with secondary IgAN have been excluded and if not, how they are distributed in the groups. It would be nice to have some information about gd-IgA to relate to the clinical course (remission and non-remission) as well as global FPE and segmental FPE.

3) Please clarify why patients with tubulointerstitial fibrosis as sign of potentially advanced IgAN with concomitant proteinuria have not been excluded

4) On which basis did you choose your remission criteria?

5) How can you explain the high percentage of MCD+ IgAN (approx. 8%) in your cohort compared to other studies ?

6) Are there patients who developed ESRD? How do you explain the eGFR decline rate of 1.17 ml/months in the segmental, non-IST group (again RAS? BP? Concomitant diseases? severe cases?)

7) How are clinical characteristics (f.e. RAS Blockade, Blood pressure, BMI, Duration of IST treatment etc.) distributed in CR, PR,NR)

Discussion:

1) Why didn’t you consider E0 as an identifier for MCD IgAN? (also in 100% of patients with global FPE)

2) Not clear if you mean your data or data in general in line170. If not your data, REF#s should be added.

3) Line 178: In my opinion a detailed discussion on the treatment needs to be done before you can conclude that resistance to IST treatment may suggest poor renal outcome- clarify also if you mean only the global FPE group or all?

4) Line180 the definition of primary and secondary response is unclear to me.

5) I would strongly suggest clarifying the discussion on adaptive changes of the podocyte and to focus on data in IgAN. The discussion about SLE and secondary IgAN in SLE is somewhat confusing, as well as the discussion about FSGS and the permeability factors (might be left out from my point of view, as you present no data on FSGS). If you included cases of secondary IgAN this might be discussed as well.

6) Line 202: Consider that gd-IgA tends to self-aggregate.

Reviewer #2: This is a well presented paper.

These data are consistent with previous nephrotic cohorts where IgA deposition in association with a podocytopathy has been shown to be steroid sensitive- whether this is a true variant of IgAN or IgAN occurring simultaneously with MCD is not known.

The authors must include the serum albumin levels of the patients in this cohort and the presence/absence of oedema to enable better interpretation of their data.

The results reinforce the 2021 KDIGO guidelines

6. PLOS authors have the option to publish the peer review history of their article (what does this mean?). If published, this will include your full peer review and any attached files.

Reviewer #1: No

Reviewer #2: **Yes: **Jonathan Barratt

---

## [Author Response · Author response to Decision Letter 0]

10 Mar 2023

Point-by-point responses to editor and reviewers (PONE-D-22-24589)

I. Need to temper the conclusion in the abstract, as the study has limitations, especially the small sample size.

=> Responses: Thanks for your valuable comment. We modified our conclusion in the abstract on page 2, lines 17-19 of the revised manuscript with track changes, as follows.

The absence of segmental sclerosis and the presence of global FPE are valuable pathological features assisting in identifying MCD-like IgAN.

II. Unclear whether a definition of MCD like IgAN would require also mesangial IgA deposition as one of the hallmarks of IgAN.

=> Responses: Thanks for your valuable comment. The diagnosis of IgAN requires predominant mesangial IgA deposits. The intensity of mesangial IgA immunostaining was required to be at least 1+ in our enrolled IgAN patients. We have added the above descriptions on page 4, lines 79-80 of the revised manuscript with track changes.

III. Need to provide all staining protocols.

=> Responses: For light microscopic examination, all specimens were fixed in buffered formalin and embedded in paraffin, and stained with hematoxylin and eosin, Masson trichrome, periodic acid-Schiff, and silver stains. For immunofluorescence staining, all frozen samples were stained with fluorescein isothiocyanate conjugated rabbit antisera to human IgG, IgA, IgM, C1q, C3, C4, kappa, and lambda light chains. For electron microscopy examination, resin-embedded samples were prepared. Mesangial electron-dense deposits and the severity of foot process effacement were assessed. We have included the above contents on page 4, lines 74-78 of the revised manuscript with track changes.

IV. Need to define in the methods which clinical and pathological features are compared in the study. 

=> Responses: Thanks for your valuable comment. We amended the descriptions in the methods on page 5, lines 88-90, and lines 108-111 of the revised manuscript with track changes, as follows.

Pathological features, including Oxford classification variables and immunostaining intensity of IgA and C3 were compared between segmental and global FPE groups. Clinical features, including hematuria, initial albumin, initial eGFR, BMI, blood pressure, nephrotic syndrome, and the presence of edema, were compared between segmental and global FPE groups.

V. Need to describe in detail the type of immunosuppression applied, and describe in detail the treatment (duration, in combination with RAS blockade etc). 

=> Responses: Thanks for your valuable comment. The immunosuppression and RAAS blockade are listed in the Table S2.

VI. Need to define over which time frame is the decline of eGFR. 

=> Responses: Thanks for your valuable comment. The baseline serum creatinine was sampled within 1 week prior to the biopsy. The time frame of eGFR decline was set within 2 years after the biopsy. We have added these descriptions on page 6, lines 122-123 of the revised manuscript with track changes.

VII. Need to mention in the Results baseline characterizes. 

=> Responses: Thanks for your valuable comment. We amended the baseline characteristics in the results on page 7, lines 161-167, as follows.

The cohort enrolled 21 males and 23 females. The mean age of the patients was 43.7 ± 17.7 years old at the time of the biopsy. Five (11.3%) had prior RAAS blockade use, and none had prior IST use. Two of the enrolled patients had diabetes mellitus. The mean systolic blood pressure was 132.8 ± 20.3 mmHg, and the mean diastolic blood pressure was 77.4 ± 13.5 mmHg. The mean BMI was 25.1 ± 4 kg/m2. Eleven (25%) of the enrolled patients presented with nephrotic syndrome. C3 immunostaining positivity was found in thirty-nine (88.6%) patients.

VIII. Need to clarify if cases with secondary IgAN have been excluded and if not, how they are distributed in the groups.

=> Responses: Thank you so much for your suggestions. After a thorough chart review of the 48 patients without segmental sclerosis, we found that two patients had ankylosing spondylitis, one had SLE, and one had Sjogren syndrome. These four patients were then excluded. Three of the four patients were not distributed in groups 1-4 since they had prior RAAS blockade use, and one was initially allocated to group 3.

The rest of the 44 patients had no history related to secondary IgAN, such as liver cirrhosis, inflammatory bowel disease, lung cancer, lymphoma, human immunodeficiency virus infection, or autoimmune diseases. We have added relevant descriptions on page 7, lines 144-149 of the revised manuscript with track changes and have revised Figures 2 and 3. Figure 4 was left unchanged since none of the four excluded patients were distributed in group 1 or group 2.

IX. Need to provide some information about gd-IgA to relate to the clinical course (remission and non-remission) as well as global FPE and segmental FPE.

=> Responses: Thanks for your valuable comment. Indeed, Gd-IgA is hypothesized to play an important role in the pathogenesis of IgAN. However, the use of Gd-IgA1 as a biomarker is not validated (Robert, 2019). Moreover, lectin-independent Gd-IgA1 ELISA is not available in our institute. We have listed this point as a limitation on page 13, lines 296-297 of the revised manuscript with track changes.

X. Unclear why patients with tubulointerstitial fibrosis as sign of potentially advanced IgAN with concomitant proteinuria have not been excluded.

=> Responses: Thanks for your valuable comment. According to the VALIGA study, all MEST variables, including tubular interstitial fibrosis associated with proteinuria at biopsy (Coppo et al., 2014), but there is no sufficient evidence to support the use of MEST variables in guiding immunosuppression therapy or not (2021). Additionally, it was observed that the combination of steroid and RAAS blockade had a benefit in renal function decline than RAAS blockade alone in IgAN classified as T1, as compared with T0 (Tesar et al., 2015).

Moreover, interstitial fibrosis is frequently associated with hypertension in adults, particularly in the elderly, and does not exclude MCD (Vivarelli et al., 2017). Therefore, we did not exclude patients classified as T1. We have added the above contents in the methods section on page 4, lines 60-65 of the revised manuscript with track changes.

XI. Need to clarify on which basis the remission criteria were chosen.

=> Responses: Thanks for your valuable comment. The remission criteria were defined by the quantitative change of proteinuria but were not used consistently. We adopted the remission criteria in the most common fashion, as mentioned in Chapter 1 of KDIGO guidance (2021). We have cited this reference and added the above contents in the methods section on page 5, lines 116-117, and page 6, lines 118-119 of the revised manuscript with track changes.

XII. Need to explain the high percentage of MCD+ IgAN (approx. 8%) in the cohort compared to other studies. 

=> Responses: Thanks for your valuable comment. Sixteen patients were classified as global FPE and accounted for 7% of 228 biopsy-proven IgAN patients. Eleven patients were classified as having nephrotic syndrome, and the percentage was 4.8%. One review article stated that nephrotic syndrome was seen in 5% of IgAN patients (Barratt and Feehally, 2005). One Korean cohort study reported that 48 among 581 IgAN patients (8.3%) presented with nephrotic syndrome (Kim et al., 2009). However, the hypoalbuminemia criteria in this Korean cohort were extended to be lower than 3.5mg/dL, which might account for the higher prevalence of nephrotic syndrome in IgAN patients. We have discussed this point in the discussion section on page 11, lines 218-224 of the revised manuscript with track changes.

XIII. Need to explain the eGFR decline rate of 1.17 ml/months in the segmental, non-IST group. 

=> Responses: Thanks for your valuable comment. After the subtraction of secondary IgAN patients and recalculation, the eGFR decline rate had no significant difference between group 3 (with IST) and group 4 (without IST). However, the eGFR decline rate seemed to deteriorate faster in group 4 than in group 3 (-0.52 [-1.72, 0.47] vs. 0.31 [-0.71, 0.80] mL/min/1.73 m2/month, p = 0.205). Clinical characteristics of each group were listed in table 4, and no significance was found between group 3 and group 4.

In an Italian randomized controlled trial, ninety-seven IgAN patients with urine protein>1g/day were enrolled, and these patients were allocated to oral steroid therapy plus ramipril or ramipril only. The follow-duration was 2-9 years. The mean rate of eGFR decline rate was faster in the ramipril-only group (-6.17 ± 13.3 vs. −0.56 ± 7.62 mL/min/1.73 m2/year; p = 0.013) (Manno et al., 2009). No such significance was observed in our cohort. Relatively short, inconsistent follow-up duration and small sample size in our cohort and higher initial prednisolone dosage (1 mg/kg/day) in that Italian study may contribute to this difference. In a retrospective analysis of the VALIGA study, the combination use of RAAS blockade and steroid was found to have a benefit in renal function decline than RAAS blockade alone, and it was observed in both patients with eGFR <50 and eGFR >50 mL/min/ 1.73 m2(Tesar et al., 2015). We have added relevant descriptions in the discussion section on page 11, lines 240-245, and page 12, lines 246-254 of the revised manuscript with track changes.

XIV. Consider a detailed discussion on the treatment before the authors can conclude that resistance to IST treatment may suggest poor renal outcome.

=> Responses: Thanks for your valuable comment. This viewpoint was concluded from one retrospective cohort of MCD mentioned in that paragraph on page 11, lines 229-234 (Szeto et al., 2015). In our study, we cannot conclude that resistance to IST in MCD-like IgAN indicates poor renal outcome. The steroid-resistant MCD-like IgAN may exhibit poor renal outcomes is an assumption that needs further investigation. We have added these descriptions in the discussion section on page 11, lines 235-239 of the revised manuscript with track changes.

XV. Need to clarify the discussion on adaptive changes of the podocyte and to focus on data in IgAN.

=> Responses: Thank you so much for your suggestions. This paragraph is meant to reason that segmental FPE is likely to be an adaptive change to IgA nephropathy (secondary), and global FPE is likely to be a coexistence of MCD (another primary disease). So we provided an analogous phenomenon seen in FSGS, lupus nephritis. For simplicity, we subtracted some paragraphs, as shown in the discussion section on page 12, lines 265-272 of the revised manuscript with track changes.

(xvi) Need to include the serum albumin levels of the patients in this cohort and the presence/absence of edema to enable better interpretation of the data.

=> Responses: Thank you so much for your suggestions. We have supplemented the information about serum albumin and the physical finding of edema in Table 1.

Reviewer #1: Wang et al report the results of an observational study aiming to evaluate key pathological features of MCD-like IgAN. The topic is relevant to further differentiate MCD like IgAN. The authors evaluated a cohort of 228 patients with biopsy proven IgAN.

The strength of the study is the effort made to further determine MCD like IgAN. Limitations are related to the small sample size, missing clinical data and definitions of treatment periods and an inconsistent discussion of the data.

I think that the study might be published only if an extensive revision of the manuscript has been performed to achieve a valid contribution to the scientific record.

Comments to the authors:

Abstract:

(1) be careful with a too confident conclusion, as the study has limitations, especially sample size. 

=> Responses: Thanks for your suggestions. We modified our conclusion in the abstract on page 2, lines 17-19 of the revised manuscript with track changes, as follows.

The absence of segmental sclerosis and the presence of global FPE are valuable pathological features that assist in identifying MCD-like IgAN.

Introduction:

(1) Proteinuria is not “frequently minimal” in IgAN and macrohematuria after upper airway infection is a classic and common in IgAN.

=> Responses: Thanks for your suggestions. We made revisions on page 3, lines 24-25 of the revised manuscript with track changes, as follows.

Asymptomatic hematuria is a common manifestation of IgAN. Concomitant minimal proteinuria may be discovered.

(2) IgA is deposited in the Mesangium rather than in the mesangial cell

=> Responses: Thanks for your suggestions. We made revisions on page 3, lines 37-38 of the revised manuscript with track changes, as follows.

Although IgAN is characterized by predominant IgA deposition in the mesangium, podocytopathy may present.

(3) Wouldn’t a definition of MCD like IgAN require also Mesangial IgA deposition as one of the hallmarks of IgAN?

=> Responses: Thanks for your valuable comment. The diagnosis of IgAN requires predominant mesangial IgA deposits. The intensity of mesangial IgA immunostaining was required to be at least 1+ in our enrolled IgAN patients. We have added the above descriptions on page 4, lines 79-80 of the revised manuscript with track changes.

Material and Methods:

(1) All staining protocols are missing, essential for reproduction of data. C3 staining is not mentioned in the methods and is found only in the supplement. IgA antibodies and procedures for IF are not described etc. They should be provided.

=> Responses: Thanks for your suggestions. For light microscopic examination, all specimens were fixed in buffered formalin and embedded in paraffin, and stained with hematoxylin and eosin, Masson trichrome, periodic acid-Schiff, and silver stains. For immunofluorescence staining, all frozen samples were stained with fluorescein isothiocyanate conjugated rabbit antisera to human IgG, IgA, IgM, C1q, C3, C4, kappa, and lambda light chains. For electron microscopy examination, resin-embedded samples were prepared. Mesangial electron-dense deposits and the severity of foot process effacement were assessed. We have included the above contents on page 4, lines 74-78 of the revised manuscript with track changes.

(2) Define in the methods which clinical and pathological features are compared in the study.

=> Responses: Thanks for your valuable comment. We amended the descriptions in the methods on page 5, lines 88-90, and lines 108-111 of the revised manuscript with track changes, as follows.

Pathological features, including Oxford classification variables and immunostaining intensity of IgA and C3, were compared between segmental and global FPE groups. Clinical features, including hematuria, initial albumin, initial eGFR, BMI, blood pressure, nephrotic syndrome, and the presence of edema, were compared between segmental and global FPE groups.

(3) Describe in detail the type of immunosuppression applied (steroid or other?) and describe in detail the treatment (duration, in combination with RAS Blockade etc.) Has RAS Blockade been applied in patients without IST (in all patients after diagnosis)?

=> Responses: Thanks for your valuable comment. The immunosuppression and RAAS blockade of all enrolled patients are listed in Table S2.

(4) Define the decline of eGFR: over which time frame?

=> Responses: Thanks for your valuable comment. The baseline serum creatinine was sampled within 1 week prior to the biopsy. The time frame of eGFR decline was set within 2 years after the biopsy. We have added these descriptions on page 6, lines 122-123 of the revised manuscript with track changes.

Results:

(1) Baseline characteristics: SBP and DBP, percentage of NS, C3 deposition, Diabetes mellitus (if present), BMI should be mentioned.

=> Responses: Thanks for your valuable comment. We amended the baseline characteristics in the results on page 7, lines 161-167, as follows.

The cohort enrolled 21 males and 23 females. The mean age of the patients was 43.7 ± 17.7 years old at the time of the biopsy. Five (11.3%) had prior RAAS blockade use, and none had prior IST use. Two of the enrolled patients had diabetes mellitus. The mean systolic blood pressure was 132.8 ± 20.3 mmHg, and the mean diastolic blood pressure was 77.4 ± 13.5 mmHg. Eleven (25%) of the enrolled patients presented with nephrotic syndrome. The mean BMI was 25.1 ± 4 kg/m2. C3 immunostaining positivity was found in thirty-nine (88.6%) patients.

(2) Please clarify if cases with secondary IgAN have been excluded and if not, how they are distributed in the groups. It would be nice to have some information about gd-IgA to relate to the clinical course (remission and non-remission) as well as global FPE and segmental FPE.

=> Responses: Thank you so much for your suggestions. After a thorough chart review of the 48 patients without segmental sclerosis, we found that two patients had ankylosing spondylitis, one had SLE, and one had Sjogren syndrome. These four patients were then excluded. Three of the four patients were not distributed in groups 1-4 since they had prior RAAS blockade use, and one was initially allocated to group 3.

The rest of the 44 patients had no history related to secondary IgAN, such as liver cirrhosis, inflammatory bowel disease, lung cancer, lymphoma, human immunodeficiency virus infection, or autoimmune diseases. We have added relevant descriptions on page 7, lines 144-149 of the revised manuscript with track changes and have revised Figures 2 and 3. Figure 4 was left unchanged since none of the four excluded patients were distributed in group 1 or group 2.

(3) Please clarify why patients with tubulointerstitial fibrosis as sign of potentially advanced IgAN with concomitant proteinuria have not been excluded

=> Responses: Thanks for your valuable comment. According to the VALIGA study, all MEST variables, including tubular interstitial fibrosis associated with proteinuria at biopsy (Coppo et al., 2014). But no sufficient evidence to support the use of MEST variables in guiding immunosuppression therapy or not (KDIGO 2021 Clinical Practice Guideline for the Management of Glomerular Diseases, 2021). Additionally, it was observed that the combination of steroid and RAAS blockade had a benefit in renal function decline than RAAS blockade alone in IgAN classified as T1, as compared with T0 (Tesar et al., 2015). 

Moreover, interstitial fibrosis is frequently associated with hypertension in adults, particularly in the elderly, and does not exclude MCD (Vivarelli et al., 2017). Therefore, we did not exclude patients classified as T1. We have added the above contents in the methods section on page 4, lines 60-65 of the revised manuscript with track changes.

(4) On which basis did you choose your remission criteria?

=> Responses: Thanks for your valuable comment. The remission criteria were defined by the quantitative change of proteinuria but were not used consistently. We adopted the remission criteria in the most common fashion, as mentioned in Chapter 1 of KDIGO guidance (2021). We have cited this reference and added the above contents in the methods section on page 5, lines 116-117, and page 6, lines 118-119 of the revised manuscript with track changes.

(5) How can you explain the high percentage of MCD+ IgAN (approx. 8%) in your cohort compared to other studies ?

=> Responses: Thanks for your valuable comment. Sixteen patients were classified as global FPE and accounted for 7% of 228 biopsy-proven IgAN patients. Eleven patients were classified as having nephrotic syndrome, and the percentage was 4.8%. One review article stated that nephrotic syndrome was seen in 5% of IgAN patients (Barratt and Feehally, 2005). One Korean cohort study reported that 48 among 581 IgAN patients (8.3%) presented with nephrotic syndrome (Kim et al., 2009). However, the hypoalbuminemia criteria in this Korean cohort were extended to be lower than 3.5mg/dL, which might account for the higher prevalence of nephrotic syndrome in IgAN patients. We have discussed this point in the discussion section on page 11, lines 218-224 of the revised manuscript with track changes.

(6) Are there patients who developed ESRD? How do you explain the eGFR decline rate of 1.17 ml/months in the segmental, non-IST group (again RAS? BP? Concomitant diseases? severe cases?)

=> Responses: Thanks for your valuable comment. One patient in group 4 received peritoneal dialysis about 1 year after the biopsy. Her initial serum creatinine was 3.89 mg/dL. Another one in group 3 reached renal endpoint (eGFR decline) within 2 years after biopsy and underwent hemodialysis 5 years after biopsy. His initial serum creatinine was 3.7mg/dL. Clinical characteristics of each group were listed in table 4, and no significance was found between group 3 and group 4.

After the subtraction of secondary IgAN patients and recalculation, the eGFR decline rate had no significant difference between group 3 (with IST) and group 4 (without IST). However, the eGFR decline rate seemed to deteriorate faster in group 4 than in group 3 (-0.52 [-1.72, 0.47] vs. 0.31 [-0.71, 0.80] mL/min/1.73 m2/month, p = 0.205).

In an Italian randomized controlled trial, ninety-seven IgAN patients with urine protein > 1 g/day were enrolled, and these patients were allocated to oral steroid therapy plus ramipril or ramipril only. The follow-duration was 2-9 years. The mean rate of eGFR decline rate was faster in the ramipril-only group (-6.17 ± 13.3 vs. −0.56 ± 7.62 mL/min/1.73 m2/year; p = 0.013) (Manno et al., 2009). No such significance was observed in our cohort. Relatively short, inconsistent follow-up duration and small sample size in our cohort and higher initial prednisolone dosage (1 mg/kg/day) in that Italian study may contribute to this difference. In a retrospective analysis of the VALIGA study, the combination use of RAAS blockade and steroid was found to have a benefit in renal function decline than RAAS blockade alone, and it was observed in both patients with eGFR <50 and eGFR >50 mL/min/ 1.73 m2 (Tesar et al., 2015). We have added relevant descriptions in the discussion section on page 11, lines 240-245, and page 12, lines 246-254 of the revised manuscript with track changes.

(7) How are clinical characteristics (f.e. RAS Blockade, Blood pressure, BMI, Duration of IST treatment etc.) distributed in CR, PR, NR)

=> Responses: Thanks for your valuable comment. Clinical characteristics according to immunosuppression therapy responsiveness are listed in Table S3.

 

Discussion:

(1) Why didn’t you consider E0 as an identifier for MCD IgAN? (also in 100% of patients with global FPE)

=> Responses: Thanks for your suggestions. The diagnostic feature of MCD includes the absence of segmental sclerosis, but the absence of endocapillary hypercellularity is not included(Fogo et al., 2015). Therefore, S = 0 by the Oxford classification is required. E = 0 is not required for enrollment, although patients in group 1 all happened to be classified as E = 0. One cohort regarding corticosteroid use in MCD-like IgAN did not include E = 0 as enrollment criteria(Wang et al., 2013). We have added relevant descriptions in the methods section on page 4, lines 55-60 of the revised manuscript with track changes

(2) Not clear if you mean your data or data in general in line 170. If not your data, REF#s should be added.

=> Responses: Thanks for your comments. The conclusion was made from the paragraph on page 11, lines 229-234 (Szeto et al., 2015). 

(3) Line 178: In my opinion a detailed discussion on the treatment needs to be done before you can conclude that resistance to IST treatment may suggest poor renal outcome- clarify also if you mean only the global FPE group or all?

=> Responses: Thanks for your valuable comment. This viewpoint was concluded from one retrospective cohort of MCD mentioned in that paragraph on page 11, lines 229-234 (Szeto et al., 2015). In our study, we cannot conclude that resistance to IST in MCD-like IgAN indicates poor renal outcome. The steroid-resistant MCD-like IgAN may exhibit poor renal outcomes is an assumption that needs further investigation. We have added these descriptions in the discussion section on page 11, lines 235-239 of the revised manuscript with track changes.

(4) Line 180 the definition of primary and secondary response is unclear to me. I would strongly suggest clarifying the discussion on adaptive changes of the podocyte and to focus on data in IgAN. The discussion about SLE and secondary IgAN in SLE is somewhat confusing, as well as the discussion about FSGS and the permeability factors (might be left out from my point of view, as you present no data on FSGS). If you included cases of secondary IgAN this might be discussed as well.

=> Responses: Thanks for your valuable comment. This paragraph is meant to reason that segmental FPE is likely to be an adaptive change to IgA nephropathy (secondary), and global FPE is likely to be a coexistence of MCD (another primary disease). So we provided an analogous phenomenon seen in FSGS, lupus nephritis. For simplicity, we subtracted some paragraphs, as shown in the discussion section on page 12, lines 265-272 of the revised manuscript with track changes.

(5) Line 202: Consider that gd-IgA tends to self-aggregate.

=> Responses: Thanks for your valuable comment. We amended relevant descriptions on page 13, lines 281-282 of the revised manuscript with track changes, as follows.

The tendency of Gd IgA to aggregate with other Gd-IgA molecules or IgG may facilitate the process of forming an immune complex(Yan et al., 2006).

Reviewer #2: 

This is a well presented paper.

=> Responses: Thank you so much for your positive response.

These data are consistent with previous nephrotic cohorts where IgA deposition in association with a podocytopathy has been shown to be steroid sensitive- whether this is a true variant of IgAN or IgAN occurring simultaneously with MCD is not known.

=> Responses: Thanks for your comment. This study aimed for early identification of MCD-like IgAN, and prompt immunosuppressant therapy would improve renal outcome in this subset of patients. 

The authors must include the serum albumin levels of the patients in this cohort and the presence/absence of oedema to enable better interpretation of their data.

=> Responses: Thanks for your valuable comment. We have added information about the presence/absence of edema and serum albumin levels in Table 1.

The results reinforce the 2021 KDIGO guidelines

=> Responses: Thank you so much for your positive response.

References:

(2021). KDIGO 2021 Clinical Practice Guideline for the Management of Glomerular Diseases. Kidney international 100, S1-s276.

Barratt, J., and Feehally, J. (2005). IgA nephropathy. Journal of the American Society of Nephrology : JASN 16, 2088-2097.

Coppo, R., Troyanov, S., Bellur, S., Cattran, D., Cook, H.T., Feehally, J., Roberts, I.S., Morando, L., Camilla, R., Tesar, V., et al. (2014). Validation of the Oxford classification of IgA nephropathy in cohorts with different presentations and treatments. Kidney international 86, 828-836.

Fogo, A.B., Lusco, M.A., Najafian, B., and Alpers, C.E. (2015). AJKD Atlas of Renal Pathology: Minimal Change Disease. American journal of kidney diseases : the official journal of the National Kidney Foundation 66, 376-377.

Kim, S.M., Moon, K.C., Oh, K.H., Joo, K.W., Kim, Y.S., Ahn, C., Han, J.S., and Kim, S. (2009). Clinicopathologic characteristics of IgA nephropathy with steroid-responsive nephrotic syndrome. Journal of Korean medical science 24 Suppl, S44-49.

Manno, C., Torres, D.D., Rossini, M., Pesce, F., and Schena, F.P. (2009). Randomized controlled clinical trial of corticosteroids plus ACE-inhibitors with long-term follow-up in proteinuric IgA nephropathy. Nephrology, dialysis, transplantation : official publication of the European Dialysis and Transplant Association - European Renal Association 24, 3694-3701.

Robert, T. (2019). The Potential Clinical Significance of the Biomarker IgA1. Kidney international reports 4, 1661-1663.

Szeto, C.C., Lai, F.M., Chow, K.M., Kwan, B.C., Kwong, V.W., Leung, C.B., and Li, P.K. (2015). Long-term outcome of biopsy-proven minimal change nephropathy in Chinese adults. American journal of kidney diseases : the official journal of the National Kidney Foundation 65, 710-718.

Tesar, V., Troyanov, S., Bellur, S., Verhave, J.C., Cook, H.T., Feehally, J., Roberts, I.S., Cattran, D., and Coppo, R. (2015). Corticosteroids in IgA Nephropathy: A Retrospective Analysis from the VALIGA Study. Journal of the American Society of Nephrology : JASN 26, 2248-2258.

Vivarelli, M., Massella, L., Ruggiero, B., and Emma, F. (2017). Minimal Change Disease. Clinical journal of the American Society of Nephrology : CJASN 12, 332-345.

Wang, J., Juan, C., Huang, Q., Zeng, C., and Liu, Z. (2013). Corticosteroid therapy in IgA nephropathy with minimal change-like lesions: a single-centre cohort study. Nephrology, dialysis, transplantation : official publication of the European Dialysis and Transplant Association - European Renal Association 28, 2339-2345.

Yan, Y., Xu, L.X., Zhang, J.J., Zhang, Y., and Zhao, M.H. (2006). Self-aggregated deglycosylated IgA1 with or without IgG were associated with the development of IgA nephropathy. Clinical and experimental immunology 144, 17-24.

---

## [Decision Letter · Decision Letter 1]

8 May 2023

PONE-D-22-24589R1Key pathological features characterize minimal change disease-like IgA nephropathyPLOS ONE

Dear Dr. Yang,

Thank you for submitting your manuscript to PLOS ONE. After careful consideration, we feel that it has merit but does not fully meet PLOS ONE’s publication criteria as it currently stands. Therefore, we invite you to submit a revised version of the manuscript that addresses the points raised during the review process.

The revised version of the manuscript is improved. However, several issues still remain to be addressed. Some of them include, i) need to acknowledge the limitations related to the small sample size of the single groups and the differing treatment protocols; ii) need further discussion of the data; iii) need to reorganize the manuscript to improve legibility and easy understanding of the issues presented; iv) need to address the remaining detailed comments of Reviewer 1 (concerning abstract, introduction, figures, tables, and discussion), all relevant to definitely improve the presentation and clarity of the manuscript.

We look forward to receiving your revised manuscript.

Kind regards,

Giuseppe Remuzzi

Academic Editor

PLOS ONE

Reviewers' comments:

Reviewer's Responses to Questions

**Comments to the Author**

1. If the authors have adequately addressed your comments raised in a previous round of review and you feel that this manuscript is now acceptable for publication, you may indicate that here to bypass the “Comments to the Author” section, enter your conflict of interest statement in the “Confidential to Editor” section, and submit your "Accept" recommendation.

Reviewer #1: (No Response)

Reviewer #2: All comments have been addressed

2. Is the manuscript technically sound, and do the data support the conclusions?

Reviewer #1: Partly

Reviewer #2: Yes

3. Has the statistical analysis been performed appropriately and rigorously? 

Reviewer #1: N/A

Reviewer #2: Yes

4. Have the authors made all data underlying the findings in their manuscript fully available?

Reviewer #1: Yes

Reviewer #2: Yes

5. Is the manuscript presented in an intelligible fashion and written in standard English?

Reviewer #1: No

Reviewer #2: Yes

6. Review Comments to the Author

Reviewer #1: Wang et al report the results of an observational study aiming to evaluate key pathological features of MCD-like IgAN. The topic is relevant to further differentiate MCD like IgAN. The authors evaluated a cohort of 228 patients with biopsy proven IgAN and found only a small number of patients that might have the characteristics of MCD-like IgAN.

The results of the study support the recommendations to treat MCD like IgAN with immunosuppression. It points also out that global podocyte food effacement (compared to segmental podocyte food effacement) is a valuable pathologic feature for MCD like IgAN.

The strength of the study is the effort made to further determine MCD like IgAN. Limitations are related to the small sample size of the single groups and the differing treatment protocols. Missing clinical data have now been added as well as partially the definitions of treatment periods. The discussion of the data needs further work. Furthermore the organization of the manuscript lacks legibility and easy understanding of the issues presented.

The authors answered many of the reviewers questions. Still the manuscript has flaws as figure legends are missing, detailed staining protocols are again not provided.

The manuscript is considered unsuitable for publication in its present form, but as much work has already been done, the authors should be encouraged to resubmit a cleaned, organized and revised version.

Comments in detail:

Abstract:

Line 5: We aimed to identify the key pathological features of MCD-like IgAN.

- that was the reason for the question, if your work would support, that also E0 could be a key pathological feature as 100% in Global FPE are E0 ( but also 88,5 in the segmental FPE Group – so the answer is probably no).

Introduction:

Line 25 : I would suggest you the following: The frequent clinical manifestations are asymptomatic hematuria and proteinuria of differing degrees.

I would also strongly suggest to give more background information on published work concerning MCD IgAN.

Line 58- Line 65: Confusing explanation and grammar. I would suggest not to add this to the manuscript. ( See also comment to line 5). The question was rather, if tubulointerstitial fibrosis interferes with proteinuria which could have an impact on your clinical parameters ( CR,PR and NR).

Line 73-113:

Describe in this paragraph the immunosuppression used for the patient groups to facilitate the understanding. IST is not only Steroids but often also CYC in your population.

Revised Figure 2: The groups that are defined in Figure2 need to be described and defined in the material and methods section. Why did you exclude 4 patients with UPCR <1 g from the segmental FPE with IST group?

Line 78: The stainings are mentioned but your staining protocols are still missing. I suggest to add these protocols to the manuscript

Line 107: better nephrotic range proteinuria

Line 116 : It is always desirable to follow the KDIGO guidelines but maybe not necessary to cite here, when you define your own remission criteria which are (to my best knowledge) not displayed 1:1 in the KDIGO Guidelines IgAN section.

Line 119: Please add: At which timepoint you decided if CR; NR oder PR were achieved. At the end of IST?

Line 166: better nephrotic range proteinuria.

Table 4 : Clinical Characteristics: Add N numbers of the groups . The legend contains CR, PR and NR which are until now not displayed in the table.

Table S3: Spironolactone is no RAS Inhibitor.

Figure 3: Where are the figure legends? P- value described in the text not in the figure.

The figures are not in the right order.

Line 195-199 Information found in table 3? Please mention in the text. Add group numbers from the text to the table 3. I.e. Group 1=Global FPE(IST+) for better and quicker understanding.

Discussion:

Line 218-224: What is your conclusion drawn from these informations?

Line 225: Not clear, you mean that the KDIGO guidelines recommend treatment with steroids?

Line 243-255: The discussion on generally treating IgAN with steroids or not does rather not apply to your study- The studies cited here did not focus on MCD like IgAN. I would suggest not to discuss that.

Line 285 maybe you could explain the importance of the former sentences for your study?

Line 293: Limitations: I would suggest to add the small group sizes, the differences in IST treatment in dose and drugs ( Group 3 did never receive CYC) and that finally an negative effect of higher blood pressure in group 3 cannot be excluded.

I would suggest not mention that you do not have the possibility to measure gd-IgA as a limitation. It would have been interesting for a discussion about IgAN activity in patients with simulanteous MCD features but is not strictly necessary.

Concerning answer 6) and the ESRD patients. It is quite surprising, that with these Creatinine values at diagnosis the patients had still an S0 . Can you comment on this?

Reviewer #2: (No Response)

7. PLOS authors have the option to publish the peer review history of their article (what does this mean?). If published, this will include your full peer review and any attached files.

Reviewer #1: No

Reviewer #2: No

---

## [Author Response · Author response to Decision Letter 1]

12 Jun 2023

Point-by-point responses to the editor and reviewers (PONE-D-22-24589R1)

i) need to acknowledge the limitations related to the small sample size of the single groups and the differing treatment protocols; 

=> Responses: Thanks for your valuable comment. We added small sample size, and variance of regimen in the part of limitation in lines 313-315 of the revised manuscript with track changes, as follows.

Third, sample size was small, owing to low prevalence of the IgAN with MCD. Lastly, different IST regimens were implemented in the treatment groups.

ii) need further discussion of the data; 

=> Responses: Thanks for your suggestion. We added relevant discussion of Oxford classification in lines 228-240 as follows.

Absence of endocapillary hypercellularity (E = 0) was not required in our cohort. Though all of the patient with global FPE happened to be classified as E = 0, the proportion had no significant difference to that of the segmental group. Additionally, one drawback of endocapillary hypercellularity scoring is that its reproducibility was poorer than segmental sclerosis scoring. Counting macrophage numbers in CD68-stained section of glomerulus may be an appropriate alternative for endocapillary hypercellularity scoring. 

Tubulointerstitial fibrosis (T = 1 or 2) is frequently associated with hypertension in adults, particularly in the elderly, and does not necessarily exclude MCD. So it was not listed as an exclusion criteria in our study. In a multi-center retrospective cohort, patients with pathological change T1 were found to be more resistant to steroid. In our cohort, three patients were resistant to steroid in treatment group and two of them were pathologically classified as T = 1. By contrast, none of the steroid-responsive IgAN patients were pathologically classified as T = 1 or 2.

iii) need to reorganize the manuscript to improve legibility and easy understanding of the issues presented; 

=> Responses: Thanks for your comment. We made some improvement to our article, listed as points below:

1. Descriptions about IST regimen, and group stratification were added so as to better understanding the article in lines 116-120 of the revised manuscript with track changes, as follows.

The IST regimen was either steroid alone, steroid in combination with cyclophosphamide, or steroid in combination with mycophenolate mofetil. Global FPE with IST, global FPE without IST, segmental FPE with IST, and segmental FPE without IST were designated as group 1, group 2, group 3, and group 4 respectively.

2. We reorganized the paragraph discussing immunosuppressive therapy in MCD-like IgAN in lines 248-269 of the revised manuscript with track changes, as follows.

KDIGO guidance recommends that IgAN with MCD-like features should be treated as MCD [1]. However, the recommendation lack valid evidence because of the rare population, and the related studies enrolled IgAN patients with nephrotic syndrome rather than IgAN with MCD-like pathological features [2-4]. In our study, IST in MCD-like IgAN patients attenuated the eGFR decline rate and had better renal outcomes than the non-IST-treated group. These patients mostly achieved CR, and the time to IST was within 3 months. Only one patient classified as MCD-like IgAN was defined as NR in treatment response. Whether the resistant response to first-line glucocorticoid treatment in MCD-like IgAN patients suggest poor renal outcomes needs further investigation. In condition of MCD, a substantial portion of patients progressed to ESKD if unresponsive to glucocorticoid treatment. 

3. We deleted the paragraph discussing immunosuppressive therapy in non MCD-like IgAN patients for simplicity in lines 280-284 of the revised manuscript with track changes.

iv) need to address the remaining detailed comments of Reviewer 1 (concerning abstract, introduction, figures, tables, and discussion), all relevant to definitely improve the presentation and clarity of the manuscript.

=> Responses: 

Thanks for your comment. We addressed comments of Reviewer 1 and the details are presented in the following section.

Reviewer #1

Abstract:

Line 5: We aimed to identify the key pathological features of MCD-like IgAN.

- that was the reason for the question, if your work would support, that also E0 could be a key pathological feature as 100% in Global FPE are E0 (but also 88.5 in the segmental FPE Group – so the answer is probably no).

=> Responses: Thanks for your comment. We added relevant descriptions in lines 228-230 of the revised manuscript with track changes, as follows. 

Absence of endocapillary hypercellularity (E = 0) was not required in our cohort. Though all of the patient with global FPE happened to be classified as E = 0, the proportion had no significant difference to that of the segmental group.

Introduction:

Line 25: I would suggest you the following: The frequent clinical manifestations are asymptomatic hematuria and proteinuria of differing degrees. I would also strongly suggest to give more background information on published work concerning MCD IgAN.

=> Responses: Thanks for your comment. We added information about MCD-like IgAN in lines 25-29 of the revised manuscript with track changes, as follows.

Nephrotic syndrome occurs in 5% of IgAN patients[5]. IgAN with nephrotic syndrome is considered to be a variant form of IgAN and can have electron microscopic features resembling minimal change disease (MCD). The treatment strategy in this MCD-like variant form is in accordance with MCD rather than IgAN.

Line 58- Line 65: Confusing explanation and grammar. I would suggest not to add this to the manuscript. (See also comment to line 5). The question was rather, if tubulointerstitial fibrosis interferes with proteinuria which could have an impact on your clinical parameters (CR, PR and NR).

=> Responses: Thanks for your suggestions. This paragraph was meant to explain why E1 and T1 were not implemented as exclusion criteria. We revised and putted this paragraph into discussion section in lines 228-240 of the revised manuscript with track changes, as follows.

Absence of endocapillary hypercellularity(E = 0) was not required in our cohort. Though all of the patient with global FPE happened to be classified as E = 0, the proportion had no significant difference to that of the segmental group. Additionally, one drawback of endocapillary hypercellularity scoring is that its reproducibility was poorer than segmental sclerosis scoring[6]. Counting macrophage numbers in CD68-stained section of glomerulus may be an appropriate alternative for endocapillary hypercellularity scoring[7]. 

Tubulointerstitial fibrosis (T = 1 or 2) is frequently associated with hypertension in adults, particularly in the elderly, and does not necessarily exclude MCD[8]. So it was not listed as an exclusion criteria in our study. In a multi-center retrospective cohort, patients with pathological change T1 were found to be more resistant to steroid[9]. In our cohort, three patients were resistant to steroid in treatment group and two of them were pathologically classified as T = 1. By contrast, none of the steroid-responsive IgAN patients were pathologically classified as T = 1 or 2.

Line 73-113:

Describe in this paragraph the immunosuppression used for the patient groups to facilitate the understanding. IST is not only Steroids but often also CYC in your population.

=> Responses: Thanks for your comment. We added relevant descriptions as below in lines 116-118 of the revised manuscript with track changes:

The IST regimen was either steroid alone, steroid in combination with cyclophosphamide, or steroid in combination with mycophenolate mofetil.

Revised Figure 2: The groups that are defined in Figure 2 need to be described and defined in the material and methods section. Why did you exclude 4 patients with UPCR <1 g from the segmental FPE with IST group?

=> Responses: Thanks for your comment. 

Global FPE with IST, global FPE without IST, segmental FPE with IST, and segmental FPE without IST were designated as group 1, group 2, group 3, and group 4 respectively. We added above descriptions in lines 118-120 of the revised manuscript with track changes.

According to KDIGO guideline, glucocorticoid treatment is not considered in classical IgAN patient with UPCR < 1g/day after maximal supportive treatment. Therefore, patients with UPCR < 1g/day in group 3 (segmental FPE with IST) were excluded.

Line 78: The stainings are mentioned but your staining protocols are still missing. I suggest to add these protocols to the manuscript

=> Responses: Thanks for your comment. The staining protocol was added in supplementary information.

Line 107: better nephrotic range proteinuria

=> Responses: Thanks for your suggestions. We revised the paragraph in lines 109-110 of the revised manuscript with track changes, as follows.

Nephrotic range proteinuria was defined as UPCR >3.5 g/g. Nephrotic syndrome was defined as nephrotic range proteinuria, serum albumin < 2.5 g/dL, and presence of edema.

Line 116: It is always desirable to follow the KDIGO guidelines but maybe not necessary to cite here, when you define your own remission criteria which are (to my best knowledge) not displayed 1:1 in the KDIGO Guidelines IgAN section.

=> Responses: Thank you so much for your suggestion. We modified this sentence in lines 123-126 of the revised manuscript with track changes, as follows.

The response to IST was defined as complete remission (CR) if UPCR < 0.3 g/g after treatment, partial remission (PR) if proteinuria reduction > 50% but UPCR > 0.3 g/g, and no response (NR) if proteinuria reduction < 50%.

Line 119: Please add: At which timepoint you decided if CR; NR oder PR were achieved. At the end of IST?

=> Responses: Thanks for your comment. 

The follow-up UPCR was assessed approximately 3 months after initiation of IST. We added above statement in lines 126-127 of the revised manuscript with track changes.

Line 166: better nephrotic range proteinuria.

=> Responses: Thanks for your suggestion. We added relevant sentence in lines 172-173 of the revised manuscript with track changes, as follows.

Fourteen (31.8%) of the enrolled patients had nephrotic range proteinuria.

Table 4: Clinical Characteristics: Add N numbers of the groups. The legend contains CR, PR and NR which are until now not displayed in the table.

Table S3: Spironolactone is no RAS Inhibitor.

=> Responses: Thanks for your valuable comment. Spironolactone was only prescribed in patient no. 7. We modified RAAS blockade use of patient no. 7 in data S5. We deleted spironolactone in the RAAS blockade column in Table S2 and Table S3. N numbers of each groups were added in Table 4 and RAAS blockade proportion was recalculated. CR, PR, an NR were removed from the figure legend since these words were placed erroneously. 

Figure 3: Where are the figure legends? P- value described in the text not in the figure.

The figures are not in the right order.

=> Responses: Thanks for your comment. We added relevant description in figure legend as below: Complete remission was achieved in 81.8% of the global FPE group and 20% in the segmental FPE group (p = 0.018).

We checked the figure order, and it was correct.

Line 195-199 Information found in table 3? Please mention in the text. Add group numbers from the text to the table 3. I.e. Group 1=Global FPE(IST+) for better and quicker understanding.

=> Responses: Thanks for your comment. We added group numbers in the table 3 and relevant descriptions in the text in lines 204-205 of the revised manuscript with track changes, as follows.

eGFR change rate was compared between groups 1 and 2, groups 3 and 4, groups 1 and 3, and groups 2 and 4, as shown in Table 3.

Discussion:

Line 218-224: What is your conclusion drawn from these informations?

=> Responses: Thanks for your comment. This paragraph is meant to compare the prevalence of nephrotic syndrome in IgAN in our cohort with previous studies and it was similar to our study.

Line 225: Not clear, you mean that the KDIGO guidelines recommend treatment with steroids?

=> Responses: Thanks for your comment. We modified the sentence in lines 248-249 of the revised manuscript with track changes, as follows.

KDIGO guidance recommends steroid treatment in IgAN with MCD-like features.

Line 243-255: The discussion on generally treating IgAN with steroids or not does rather not apply to your study- The studies cited here did not focus on MCD like IgAN. I would suggest not to discuss that.

=> Responses: Thanks for your comment. The two studies cited in this paragraph found that immunosuppression therapy ameliorate eGFR decline rate in IgAN patients with no MCD-like features. In our cohort, patients in group 3 (segmental FPE with IST) appeared to have slower eGFR decline rate than group 4(segmental FPE without IST), though it had no significant statistical difference. For simplicity, we subtracted the paragraph in lines 280-284 of the revised manuscript with track changes.

Line 285 maybe you could explain the importance of the former sentences for your study?

=> Responses: Thanks for your comment. We considered that the MCD-like features in IgAN was more likely to be coexistence of MCD rather than podocytopathic variant of IgAN, as discussed in lines 285-309 of the revised manuscript with track changes. 

Line 293: Limitations: I would suggest to add the small group sizes, the differences in IST treatment in dose and drugs (Group 3 did never receive CYC) and that finally an negative effect of higher blood pressure in group 3 cannot be excluded.

=> Responses: Thanks for your comment. We added small sample size, and variance of regimen in the part of limitation in lines 313-315 of the revised manuscript with track changes, as below.

Third, sample size was small, owing to low prevalence of the IgAN with MCD. Lastly, different IST regimens were implemented in the treatment groups.

I would suggest not mention that you do not have the possibility to measure gd-IgA as a limitation. It would have been interesting for a discussion about IgAN activity in patients with simultaneous MCD features but is not strictly necessary.

=> Responses: Thank you so much for your suggestion. We deleted this description.

Concerning answer 6) and the ESRD patients. It is quite surprising, that with these Creatinine values at diagnosis the patients had still an S0. Can you comment on this?

=> Responses: Thanks for your comment. In IgAN patients with eGFR <30ml/min/1.73m2, approximately 8% was pathologically classified as S0[10]. It is possible for a IgAN patient with advanced renal impairment to be classified as S0.

Reviewer #2: (No Response)

References:

1. KDIGO 2021 Clinical Practice Guideline for the Management of Glomerular Diseases. Kidney Int. 2021;100(4s):S1-s276. Epub 2021/09/25. doi: 10.1016/j.kint.2021.05.021. PubMed PMID: 34556256.

2. Pattrapornpisut P, Avila-Casado C, Reich HN. IgA Nephropathy: Core Curriculum 2021. Am J Kidney Dis. 2021;78(3):429-41. Epub 2021/07/13. doi: 10.1053/j.ajkd.2021.01.024. PubMed PMID: 34247883.

3. Lai KN, Lai FM, Ho CP, Chan KW. Corticosteroid therapy in IgA nephropathy with nephrotic syndrome: a long-term controlled trial. Clin Nephrol. 1986;26(4):174-80. Epub 1986/10/01. PubMed PMID: 3536231.

4. Kim SM, Moon KC, Oh KH, Joo KW, Kim YS, Ahn C, et al. Clinicopathologic characteristics of IgA nephropathy with steroid-responsive nephrotic syndrome. J Korean Med Sci. 2009;24 Suppl(Suppl 1):S44-9. Epub 2009/02/12. doi: 10.3346/jkms.2009.24.S1.S44. PubMed PMID: 19194561; PubMed Central PMCID: PMCPMC2633194.

5. Barratt J, Feehally J. IgA nephropathy. J Am Soc Nephrol. 2005;16(7):2088-97. Epub 2005/06/03. doi: 10.1681/asn.2005020134. PubMed PMID: 15930092.

6. Roberts IS, Cook HT, Troyanov S, Alpers CE, Amore A, Barratt J, et al. The Oxford classification of IgA nephropathy: pathology definitions, correlations, and reproducibility. Kidney Int. 2009;76(5):546-56. Epub 2009/07/03. doi: 10.1038/ki.2009.168. PubMed PMID: 19571790.

7. Soares MF, Genitsch V, Chakera A, Smith A, MacEwen C, Bellur SS, et al. Relationship between renal CD68(+) infiltrates and the Oxford Classification of IgA nephropathy. Histopathology. 2019;74(4):629-37. Epub 2018/10/12. doi: 10.1111/his.13768. PubMed PMID: 30303541.

8. Vivarelli M, Massella L, Ruggiero B, Emma F. Minimal Change Disease. Clin J Am Soc Nephrol. 2017;12(2):332-45. Epub 2016/12/13. doi: 10.2215/cjn.05000516. PubMed PMID: 27940460; PubMed Central PMCID: PMCPMC5293332.

9. Yang P, Chen X, Zeng L, Hao H, Xu G. The response of the Oxford classification to steroid in IgA nephropathy: a systematic review and meta-analysis. Oncotarget. 2017;8(35):59748-56. Epub 2017/09/25. doi: 10.18632/oncotarget.19574. PubMed PMID: 28938678; PubMed Central PMCID: PMCPMC5601774.

10. Coppo R, Troyanov S, Bellur S, Cattran D, Cook HT, Feehally J, et al. Validation of the Oxford classification of IgA nephropathy in cohorts with different presentations and treatments. Kidney Int. 2014;86(4):828-36. Epub 2014/04/04. doi: 10.1038/ki.2014.63. PubMed PMID: 24694989; PubMed Central PMCID: PMCPMC4184028.

---

## [Decision Letter · Decision Letter 2]

26 Jun 2023

Key pathological features characterize minimal change disease-like IgA nephropathy

PONE-D-22-24589R2

Dear Dr. Yang,

We’re pleased to inform you that your manuscript has been judged scientifically suitable for publication and will be formally accepted for publication once it meets all outstanding technical requirements.

**The re-revised manuscript is further improved. The authors have now adequately addressed all the remaining comments/critiques raised by the reviewers. Thus, the manuscript is suitable for publication in PLOS ONE.**

Kind regards,

Giuseppe Remuzzi

Academic Editor

PLOS ONE

Additional Editor Comments (optional):

Reviewers' comments:

Reviewer's Responses to Questions

**Comments to the Author**

1. If the authors have adequately addressed your comments raised in a previous round of review and you feel that this manuscript is now acceptable for publication, you may indicate that here to bypass the “Comments to the Author” section, enter your conflict of interest statement in the “Confidential to Editor” section, and submit your "Accept" recommendation.

Reviewer #1: All comments have been addressed

2. Is the manuscript technically sound, and do the data support the conclusions?

Reviewer #1: Yes

3. Has the statistical analysis been performed appropriately and rigorously? 

Reviewer #1: Yes

4. Have the authors made all data underlying the findings in their manuscript fully available?

Reviewer #1: Yes

5. Is the manuscript presented in an intelligible fashion and written in standard English?

Reviewer #1: Yes

6. Review Comments to the Author

Reviewer #1: (No Response)

7. PLOS authors have the option to publish the peer review history of their article (what does this mean?). If published, this will include your full peer review and any attached files.

Reviewer #1: No

---

## [Editor Report · Acceptance letter]

12 Jul 2023

PONE-D-22-24589R2 

Key pathological features characterize minimal change disease-like IgA nephropathy 

Dear Dr. Yang:

I'm pleased to inform you that your manuscript has been deemed suitable for publication in PLOS ONE. Congratulations! Your manuscript is now with our production department. 

Kind regards, 

on behalf of

Prof. Giuseppe Remuzzi 

Academic Editor

PLOS ONE